# Immune-Related Adverse Events Due to Cancer Immunotherapy: Immune Mechanisms and Clinical Manifestations

**DOI:** 10.3390/cancers16071440

**Published:** 2024-04-08

**Authors:** Silvia Casagrande, Giulia Boscato Sopetto, Giovanni Bertalot, Roberto Bortolotti, Vito Racanelli, Orazio Caffo, Bruno Giometto, Alvise Berti, Antonello Veccia

**Affiliations:** 1Unit of Neurology, Rovereto Hospital, Azienda Provinciale per i Servizi Sanitari-APSS, 38122 Trento, Italy; silvia.casagrande@apss.tn.it (S.C.); bruno.giometto@apss.tn.it (B.G.); 2Department of Cellular, Computational and Integrative Biology (CIBIO), University of Trento, 38122 Trento, Italy; g.boscatosopetto@unitn.it (G.B.S.); giovanni.bertalot@apss.tn.it (G.B.); vito.racanelli@apss.tn.it (V.R.); 3Center for Medical Sciences (CISMed), University of Trento, 38122 Trento, Italy; 4Multizonal Unit of Pathology, APSS, 38122 Trento, Italy; 5Unit of Rheumatology, Santa Chiara Regional Hospital, APSS, 38122 Trento, Italy; roberto.bortolotti@apss.tn.it; 6Unit of Internal Medicine, Santa Chiara Regional Hospital, APSS, 38122 Trento, Italy; 7Unit of Oncology, Santa Chiara Regional Hospital, APSS, 38122 Trento, Italy; orazio.caffo@apss.tn.it (O.C.); antonello.veccia@apss.tn.it (A.V.); 8Department of Psychology and Cognitive Sciences (DIPSCO), University of Trento, 38122 Trento, Italy

**Keywords:** immune checkpoint inhibitors, ICI, immune-related adverse events, irAEs, anti-PD1/PD-L1, anti-CTLA4

## Abstract

**Simple Summary:**

This review comprehensively summarizes the pathogenic mechanisms responsible for immune-related Adverse Events (irAEs) arising from self-tolerance loss due to Immune Checkpoint Inhibitors (ICIs) and discusses the main clinical manifestations due to irAEs categorized by organ types, their incidence, and risk factors. In addition, it focuses on the different distributions in frequencies of the diverse clinical manifestations due to irAEs between CTLA4 and PD1 pathway inhibitors and the intricate differential with primary autoimmune disorders, which share some pathophysiological and clinical aspects.

**Abstract:**

The landscape of cancer treatment has undergone a significant transformation with the introduction of Immune Checkpoint Inhibitors (ICIs). Patients undergoing these treatments often report prolonged clinical and radiological responses, albeit with a potential risk of developing immune-related adverse events (irAEs). Here, we reviewed and discussed the mechanisms of action of ICIs and their pivotal role in regulating the immune system to enhance the anti-tumor immune response. We scrutinized the intricate pathogenic mechanisms responsible for irAEs, arising from the evasion of self-tolerance checkpoints due to drug-induced immune modulation. We also summarized the main clinical manifestations due to irAEs categorized by organ types, detailing their incidence and associated risk factors. The occurrence of irAEs is more frequent when ICIs are combined; with neurological, cardiovascular, hematological, and rheumatic irAEs more commonly linked to PD1/PD-L1 inhibitors and cutaneous and gastrointestinal irAEs more prevalent with CTLA4 inhibitors. Due to the often-nonspecific signs and symptoms, the diagnosis of irAEs (especially for those rare ones) can be challenging. The differential with primary autoimmune disorders becomes sometimes intricate, given the clinical and pathophysiological similarities. In conclusion, considering the escalating use of ICIs, this area of research necessitates additional clinical studies and practical insights, especially the development of biomarkers for predicting immune toxicities. In addition, there is a need for heightened education for both clinicians and patients to enhance understanding and awareness.

## 1. Introduction

In the past few years, impressive breakthroughs have been made in the field of cancer immunotherapy, which have led to a significant increase in survival outcomes for cancer patients [1,2,3]. In this context, various types of monoclonal antibodies (Abs) called immune checkpoint inhibitors (ICIs) have been developed and subsequently approved for clinical practice since 2011 [4,5].

The introduction of ICIs into clinical practice, alone or in combination with other anticancer drugs (such as chemotherapeutic or targeted agents), has dramatically changed the therapeutic approach for almost all types of cancer. Their use has become so widespread in first or subsequent lines of therapy that most cancer types now have at least one ICI in their therapeutic armamentarium (Table 1). On the other hand, clinicians quite frequently face immune-related adverse events (irAEs), requiring prompt identification and proper treatment to avoid complications and morbidity [6,7,8]. The irAEs most frequently reported are heterogeneous and can occur in various organs, including the skin (manifesting as rash), gastrointestinal system (leading to symptoms like diarrhea, colitis, and hepatitis), endocrine system (resulting in thyroiditis, hypothyroidism, and hyperthyroidism), and lungs (causing pneumonitis and interstitial lung disease) [9].

From a biological perspective, ICIs function by awakening the immune system and boosting its activation to enhance the anti-tumoral immune response. This is achieved by blocking natural immune checkpoints, which comprise a plethora of inhibitory receptors that are crucial in regulating the delicate balance of adaptive immune system activation toward non-self-antigens and maintaining self-tolerance [4,5]. Current ICIs act primarily by blocking several inhibitory receptors that function as breaks for the adaptive immune response, allowing effector T cells and other adaptive immune cells to fully activate and exert their functions.

In this review, we focused on the immune mechanisms and clinical manifestations of irAEs due to ICIs, summarizing the known pathogenic mechanisms underlying these conditions and reporting the most common clinical manifestations that clinicians should know.

## 2. Mechanisms and Targets of Immune Checkpoint Inhibitors

Within the tumor microenvironment (TME), cancer cells release tumor-specific antigens, which are processed by antigen-presenting cells (APCs), exposed in the context of major-histocompatibility complex II (MHC-II) and presented to T cells via a direct interaction with the T-cell receptor (TCR). This results in the activation of tumor antigen-specific T cells and initiation of the overall immune response against the cancer, which is usually referred to as “tumor immune surveillance”. However, cancer exploits several mechanisms to escape this anti-tumor immune response, including but not limited to activation of inhibitory signals of immune checkpoints. The action of ICIs is exploited by removing these inhibitory signals of T cell activation, stimulating tumor-reactive T cells to mount an effective antitumor response. ICIs block the inhibitory receptors, cytotoxic T lymphocyte-associated protein 4 (CTLA4) (e.g., ipilimumab) and the PD1 axis, either the programmed cell death protein 1 (PD1) (e.g., nivolumab, pembrolizumab, cemiplimab) or its ligand programmed cell death ligand 1 (PD-L1) (atezolizumab, avelumab, durvalumab) [10,11].

CTLA4 is an immunoglobulin cell surface receptor expressed on naïve clusters of differentiation (CD) 4 and CD8 T cells after stimulation [12]. CTLA4 represents a critical immune checkpoint, given its role in the early stages of T-cell development. In fact, T cell activation not only requires TCR engagement with an antigen presented by APCs in the context of MHC-II but also require a costimulatory signal, which is mediated by the binding of CD28 to the T-cell co-receptor CD80/86 on APCs. This secondary signal is essential for maintaining T-cell homeostasis, as its absence can lead to the recognition of the antigen as “self” and the development of tolerance [13].

CTLA4 acts as a strong competitive CD28 homolog that binds CD28 ligands, CD80/86, preventing downstream T cell activation. It has higher affinity for CD80 (B7-1) and, to a lesser extent, for CD86 (B7-2), compared to CD28 [14]. By binding to CD80/86, CTLA4 triggers an intracellular inhibitory feedback pathway mediated by the tyrosine phosphatase SHP-2 and the serine/threonine phosphatase PP2A, which dephosphorylate and inactivate downstream signaling kinases of the PI3K and AKT pathways [13]. Additionally, CTLA4 can act extracellularly to remove CD80/86 via trans endocytosis, further preventing T cell activation [15].

Even though physiologically CTLA4 modulates T cell priming and activation in regional secondary lymphoid organs, where it has critical functions in maintaining self-tolerance, CTLA4 is also thought to exert cell extrinsic functions in the periphery involving FoxP3+ Tregs [16,17,18]. In fact, CTLA4 is constitutively expressed on regulatory CD4+ T cells (Tregs), as it is a direct target gene of the fork-head box P3 (FoxP3) transcription factor, whose expression determines the Treg cell lineage [15]. The inhibition of CTLA4 in Tregs promotes their immunosuppressive activity and greatly impairs their ability to control the autoimmune and antitumor response [19]. Therefore, blocking CTLA4 by specific monoclonal Abs, such as ipilimumab, not only upregulates helper T-cell functions but also contributes to reducing the immunosuppression mediated by Tregs in the TME [20].

The roles of CTLA-4 and PD-1, i.e., the other immune checkpoints of T cells, are largely distinct in modulating the immune response. While CTLA-4 regulates T-cell proliferation early (acting primarily in lymph nodes), PD-1 suppresses T cells later on (primarily acting in peripheral tissues). The PD1/PD-L1 axis is responsible for maintaining peripheral tolerance through the promotion of apoptosis of antigen-specific T cells and the reduction of apoptosis of Tregs cells [21,22,23]. PD1 is also part of the immunoglobulin cell surface receptor superfamily, and it contributes to immune homeostasis by binding to its ligands PD-L1/2. PD1 expression is broader than CTLA4, as it can be found on T cells, B cells, NK cells, dendritic cells, macrophages, peripheral tissues, and also non-hematopoietic cells and non-lymphoid tissues (including tumor cells or stromal components of the TME) (Figure 1) [13,24]. PDL2 expression is more restricted, even though it has been described on some tumors, including melanoma [25]. Biologically, the expression of PD-L1 on the cancer tissues contributes to the regulation of tumorigenesis, and from a clinical standpoint, the percentage of expression of PD-L1 is measured on tumor biopsy, as shown in Figure 1, to predict the patient response to anti-PD1 treatment. While anti-PD1 therapy is not contraindicated in patients bearing tumors with low expression of PD1, tumors that express high amounts of PD-L1 on the cell membrane (>50%) may respond particularly well.

The expression of PD1 is a consequence of T- and B-cell activation, and it is mediated by the action of inflammatory cytokines, such as IFNγ, in the context of T-cell effector activities [22,26]. The binding of PD1 on T cells to its ligands on APCs triggers a negative signaling pathway mediated by the tyrosine phosphatase SHP2, which dephosphorylates the proximal signaling mediators ZAP-70 and PI3K [27]. This results in downregulation of the TCR signaling, thus switching off the process of T cell activation, proliferation, and cytokine release [28]. As PD-L1 is also expressed on cancer cells, the tumor exploits this negative feedback mechanism to evade immune surveillance. Consequently, by impeding the binding of PD-L1 to the PD1 receptor with therapeutic monoclonal Abs, the negative regulation of the immune response can be unlocked, thus enabling the initiation of an antitumor immune response. PD-L1 in tumor cells is usually detected with immunohistochemistry (Figure 1) to drive the choice of the appropriate ICI [29].

## 3. Pathogenic Mechanisms Underlying irAEs

The immune modulation resulting from ICI usage can alter self-tolerance, and it is often associated with a spectrum of autoimmune-mediated toxicities that are collectively known as immune-related Adverse Events (irAEs). According to different studies, these irAEs have been reported in up to 37% of patients in clinical trials [24,30]. The pleiotropic nature of irAEs results in the affection of many organs (e.g., skin, colon, endocrine organs, joints, heart, lungs, and the musculoskeletal system), with mechanisms that vary greatly depending on the distinct checkpoint blockers [31,32]. Due to the high heterogeneity and complexity of irAEs’ manifestation, they are often considered the “Achilles’ heel” of ICIs (Figure 2) [33]. As the applications of ICIs are increasing in clinical practice, a more comprehensive understanding of the pathophysiological mechanisms underlying irAEs is emerging as a clinical need. There is growing evidence suggesting that irAEs share mechanistic and clinical similarities with autoimmune diseases, possibly implying that irAEs could represent forms of autoimmune reactions in ICI-treated patients. In fact, irAEs arise from the overactivation of the immune system induced by ICIs, which compromise immunogenic self-tolerance and enhance the progression of autoimmune events at different immunopathogenic levels [34].

This happens with various mechanisms, including: (1) the effect of autoreactive CD4+ T cells generated during ICI therapy as a result of diversification and expansion of autoreactive cells escaping central tolerance; (2) the activation of cytotoxic CD8+ T cells against healthy tissues as a consequence of diversification and expansion of autoreactive clones; (3) the release of self-antigens from tumor cells undergoing cytotoxic attack resulting in antigenic cross-presentation and cross-reactivity to target tissues; (4) the proliferation and activation of autoreactive B cells, produced similarly to T cells, as a consequence of massive expansion, which result in the secretion of Abs mediating cell cytotoxicity in target tissues by mechanisms activating the classic complement cascade; (5) the abnormal release of pro-inflammatory mediators, including cytokines and chemokines, which decrease the function and survival of Treg cells and generate systemic and organ-specific inflammation by binding to immune cells and triggering pro-inflammatory intracellular signaling; (6) the direct off-target effect of ICIs on cells bearing the target immune checkpoint ligand, as in the case of hypophysitis developing as a consequence of anti-CTLA4 treatment (Figure 2) [35]. It is also postulated that the insurgence of irAEs could be influenced by the environment, the gut microbiome, and the host genetic background [36,37,38,39,40].

### 3.1. T-Cell-Mediated Mechanisms

Since ICIs act mainly at the levels of T cells’ immune checkpoints—increasing T cells’ activity—T-cell-mediated mechanisms are thought to be the predominant effectors of irAEs. The fundamental role of checkpoints in T cells and immune homeostasis was highlighted by genetic loss-of-function studies in mice, demonstrating that the lack of CTLA4 results in the development of lymphoproliferative disease, multiorgan lymphocytic infiltration, and tissue destruction, which rapidly lead to death due to severe myocarditis and pancreatitis [41,42]. In this context, different mechanisms have been described, including the expansion of autoreactive CD4+ and CD8+ T cell clones, cross-reactivity due to shared antigens, and Tregs dysregulation.

#### 3.1.1. Breach of Self-Tolerance of T Cells

The loss of T cell tolerance and the subsequent expansion of autoreactive T cell clones are among the mechanisms leading to autoimmune adverse events. T-cell-mediated autoimmunity associated with the loss of immune tolerance to cross-reactive antigens, in the context of ICI usage, is still a largely under-investigated area. Physiologically, a small proportion of silenced yet potentially autoreactive T cells that escaped central tolerance during their central maturation recirculate in blood and secondary lymphoid organs, remaining vulnerable to autoimmune activation by the encounter with autoantigens in the tissues. In a physiological context, peripheral immune tolerance checkpoints monitor this autoreactive status by modulating and inhibiting TCR signaling via co-receptors and their ligands [35]. It is plausible that the excessive stimulation of TCR signaling associated with ICI usage interferes with the maintenance of peripheral immune tolerance and pushes the activation of potentially autoreactive T cell clones toward autoantigens [43].

The use of ICIs is also often associated with augmented diversification in the T-cell repertoire, pointing toward the expansion of autoreactive T cell clones. In a preliminary study conducted in 2017, it was observed that the administration of the anti-CTLA4 Ab ipilimumab to cancer patients led to a pronounced diversification of both CD4+ and CD8+ T-cell repertoires. This diversification resulted in the generation of self-reactive clones within two weeks of treatment, occurring prior to the onset of clinically manifested irAEs [44]. This was also supported among all papers by a study involving metastatic prostate cancer patients who developed irAEs following treatment with ipilimumab. In this case, the CTLA4 blockade resulted in a greater diversification of TCR repertoire and an increased number of T cell clones, which preceded the onset of irAEs [44]. CTLA4-based ICIs can also lead to the expansion of the number of TCR CDR3 sequences, as reported in a study involving metastatic melanoma patients treated with tremelimumab. Interestingly, the study evidenced how patients were more likely to develop toxicities if they expanded the number of unique productive sequences, which may reflect the expansion of autoreactive T cell clones that mediate irAEs [45].

#### 3.1.2. Shared Antigens/Cross-Reactivity

The expansion of autoreactive clones is also responsible for the initiation of specific autoimmune responses mediated by those T cell clones recognizing shared antigens in normal and tumoral tissues. In fact, activated autoreactive T cells are capable of targeting normal and tumoral tissues, thus leading to the anti-tumoral response along with ICI-related toxicities, such as the development of vitiligo in melanoma patients treated with ICIs [46,47]. This mechanism was first hypothesized in a case report of fatal myocarditis in a patient treated with ipilimumab and nivolumab for advanced melanoma. Interestingly, the autopsy revealed infiltration of T cells in the myocardial tissue; additionally, TCR sequencing of the infiltrating cells revealed common high-frequency TCRs in the tumor, myocardial, and skeletal muscle, evidencing the presence of the same T cell clones in these sites and suggesting the presence of shared antigens between these tissues [48]. This was also supported by a study including metastatic non-small cell lung cancer (NSCLC) patients who developed skin ICI-related toxicities following treatment with PD1 inhibitors. Identical TCR sequences were found upon analyzing patients’ matched biopsies from tumors and skin lesions, suggesting infiltration of the same T cell clones in both sites. It was demonstrated that these clones recognized nine potential shared T cell antigens, which were specifically able to provoke IFN-γ-positive CD8+ T cell responses [47,49].

#### 3.1.3. Imbalance between Tregs and T-Effector Cells

The breach in peripheral tolerance due to an imbalance of Treg cells is another postulated mechanism for irAE development. Tregs (Foxp3+ CD25+ CD4+) have a crucial role in the maintenance of immune homeostasis, acting as gatekeepers of peripheral tolerance by eliminating autoreactive T cells [50]. The importance of Tregs is supported by the fact that patients experiencing severe autoimmunity often possess loss-of-function mutations in the Foxp3 gene [51]. This is endorsed by studies in mouse models, in which dysregulation of Tregs has been associated with severe multiorgan autoimmunity [50].

Tregs are also considered fundamental therapeutic targets in cancer immunotherapy, as they can switch the TME toward an immunosuppressive status, which results in the immune evasion of cancer cells [52,53]. The use of ICIs can directly influence and target Tregs, as CTLA4 is constitutively expressed on naïve Tregs and at higher levels on effector Tregs and on tumor-infiltrating Tregs. Hence, ICIs, especially CTLA4-targeting monoclonal Abs, lead to a fundamental depletion of Tregs. This shifts the balance in favor of effector T cells and results in the loss of peripheral tolerance and the induction of irAEs [47].

### 3.2. B-Cell-Mediated Mechanisms

In addition to the conspicuous involvement of T cells in the pathogenesis of irAEs, certain complications arising from checkpoint blockade can be ascribed to mechanisms mediated by B cells and autoAbs [54,55]. Indeed, there is a growing acknowledgment of the pivotal role in cancer immunosurveillance played by mature B cells in the TME. The role of B cells as key elements in orchestrating both the antitumoral response and the toxicities of ICIs stems from the fact that B cell biology is tightly regulated by CTLA4 and PD1 [56,57]. In fact, the impact of CTLA4 on human B cells was demonstrated by studies involving individuals with CTLA4 haploinsufficiency. These subjects exhibited a consistent reduction in the count of circulating B cells, along with an increase in the CD21^low^ subset. Notably, this evidence mimics the changes in B cells observed in cancer patients undergoing combination checkpoint blockade therapy [56,58]. In this context, Das et al. demonstrated that melanoma patients had a decline in the overall number of circulating B cells after combined ICI treatments (i.e., anti-CTLA4 and anti PD1/PD-L1), along with an increase in CD21low B cells and plasmablasts, which also correlated with an increase in the risk of irAE development [59]. Interestingly, studies on phenotype and genomics signatures of CD21low B cells reported an increased expression of autoreactive BCR and an evident resemblance with T-box transcription factor-positive (Tbet+) B cells, knowingly associated with autoimmunity [60].

Additionally, there is evidence that ICI treatment can directly modulate B cells and plasmablasts in the TME. Indeed, a study demonstrated that tumor-associated B cells express T cell and macrophage chemoattractant mediators (CCL3, CCL3L1, CCL4, CCL5, CCL28, and CXCL16), which is indicative of a primary role of B cells in orchestrating the immune response in the TME by sustaining inflammation and enhancing the recruitment and activation of other immune cells [61,62].

### 3.3. AutoAbs’ Action

The activation of autoreactive B and T cells due to ICIs leads to the production of Abs against self-antigens. AutoAbs induce irAEs, triggering the development of the autoimmune disorder by binding to their target(s) [63]. This is the case of ICI-induced thyroid dysfunction, which was observed in patients with higher titers of anti-thyroid autoAbs [64]. Additionally, autoAbs contribute to irAEs-mediated tissue damage by directly triggering the activation of the complement cascade and the formation and deposition of immune complexes in the target tissues [63]. Abs have been found to cross-reactively recognize tumor and self-antigens, which include a series of aberrantly expressed antigens, intracellular molecules, and neoantigens [65]. The serum levels of such Abs have been proposed not only as biomarkers for cancer detection and prognosis but also for ICIs’ responses and toxicities [66,67].

It has also been postulated that the pre-existence of autoAbs, a hallmark of many autoimmune diseases, in cancer patients may increase the propensity of these patients to experience autoimmune disease flares or irAEs [68,69]. This was evidenced in a systematic review involving cancer patients with pre-existing autoimmune diseases, treated with ICIs; in 75% of cases, patients experienced exacerbation of the pre-existing autoimmunity, development of de novo irAEs, or both [69]. In fact, the presence of Anti-thyroglobulin (TG) and/or anti-thyroid peroxidase (TPO) autoAbs increase the risk of ICI-induced thyroid dysfunction; the presence of anti-thyroid autoAbs has been described in 13–70% of patients who develop ICI-associated thyroid dysfunction [70,71,72,73]. Additionally, β-cell autoAbs can be found in approximately 50% of ICI-induced diabetes patients, and around 50% of patients diagnosed with ICI-induced myasthenia gravis have high levels of anti-acetylcholine receptor (AchR) Abs [33,74,75]. Additionally, autoAbs against BP180 can be found in the majority of anti-PD-L1-associated bullous pemphigoid patients. Negative rheumatoid factor (RF) and cyclic citrullinated peptide (CPP) autoAbs are often associated with ICI-induced arthritis [76,77].

### 3.4. Direct Effect of Monoclonal Abs

Another described mechanism involved in irAE development is the direct effect of ICIs on organs. As ICIs target molecules that are potentially expressed by both the immune system and normal tissues, it is plausible that some irAEs can be derived from complement-mediated tissue damage due to an “off-target” binding of ICIs to cells of certain tissues.

One of the most prominent cases is ICI-mediated hypophysitis, which is triggered by ipilimumab, as a consequence of the strong expression of CTLA4 in the anterior pituitary gland [78]. CTLA 4 has been demonstrated to be expressed in murine and human pituitary glands, hypothesizing the possibility of a direct action of ipilimumab leading to complement activation and subsequent destruction of endocrine cells [78].

The presence of anti-pituitary, anti-GNAL (guanine nucleotide-binding protein G subunit alpha), and anti-ITM2B (integral membrane protein 2B) autoAbs has been associated with ICI-induced hypophysitis [79]. It has been proposed that ICI-induced pituitary destruction results from the activation of the classic complement cascade, which is triggered by the combined action of these endogenous autoAbs and the exogenous anti-CTLA4 IgG1 Abs [78]. Pituitary toxicity perfectly reflects the complexity and multifactorial nature of irAEs [80].

### 3.5. Inflammation and Cytokine-Mediated Mechanisms

Several cytokines and chemokines are thought to play a role in the onset of irAEs, and they are collectively responsible for the development of the cytokine-release syndrome (CRS).

This type of toxicity represents a hyperinflammatory status resulting from massive cytokine release, which is postulated to be either a direct consequence of T/B cells’ immune responses or an indirect causative mechanism itself [47]. This condition was first described in the early 1990s as a consequence of anti-T-cell antibody muromonab-CD3 treatment for solid organ transplantation [81]. CRS can present either as a flu-like syndrome or as a life-threatening systemic disease defined by hypotension, capillary leakage, disseminated intravascular coagulopathy, and multi-organ failure [82].

Even though CRS can be a consequence of ICI use, its appearance in the context of irAEs is fortunately rare, with an estimated incidence of 0.07% [83].

High circulating levels of cytokines have been proposed to correlate with the onset of irAEs. In fact, increased baseline IL-17 concentrations were frequently associated with ICI-mediated colitis and severe diarrhea in melanoma patients treated with ipilimumab; increased IL-6 and IL-10 concentrations were found in patients presenting skin irAEs; increased concentrations of IL-1β, IL-2, and GM-CSF have been associated with ICI-mediated thyroid dysfunctions [84,85,86]. Interestingly, a study investigating large panels of circulating cytokines as irAEs’ prediction factors demonstrated that eleven cytokines (G-CSF, GM-CSF, fractalkine, FGF-2, IFN-α2, IL-12p70, IL-1α, IL-1β, IL-1RA, IL-2, and IL-13) were strongly associated with an increased risk of developing irAEs in melanoma patients treated with ICIs [84,87]. Another independent study investigating the changes in a panel of 40 cytokines/chemokines before and after anti-PD1 treatment in lung cancer patients reported an association between decreased baseline levels of CXCL9, CXCL10, CXCL11, and CXCL13 and irAE development [88]. These chemokines are chemotactic for activated T cells, and they belong to an important axis that regulates the differentiation of T cells toward the Th1 phenotype and may contribute to the migration of mature T cells to the tumor site.

As the above-mentioned molecules are often implicated in the inflammatory processes of autoimmune diseases, it has been postulated that a pre-existing status of inflammation characterized by elevated levels of signature cytokines could be exacerbated by ICIs and induce irAEs [89].

## 4. Host-Specific and Environmental Factors Predisposed to irAEs

Often, ICI-related toxicities are difficult to diagnose in early stages due to insidious symptoms. In this scenario, the early identification of patients who are more likely to develop irAEs before the full-blown clinical manifestations is of crucial relevance to prevent severe toxicities and treatment discontinuation [89]. The identification of predictive biomarkers and predisposing factors will help in this task. Both host-specific factors (such as genetic background and environmental factors) and commensal organisms can influence all the above-mentioned mechanisms of toxicities, from antigen-recognition by immune cells to the orchestration of the whole immune response [90]. Additionally, pre-existing autoimmune disorders have been linked to an increased risk of disease flares and/or high severity irAEs; hence, the presence of a pre-existing inflammatory status could represent a risk factor. Overall, the roles of genetics and epigenetics, the environment and the microbiota, and the underlying immune status of a patient in the predisposition to irAE development remain to be fully addressed.

### 4.1. Role of the Host Microbiota

One of the most captivating research fields in the context of ICIs relates to the influence of the host bacterial composition in shaping both the efficacy and the toxicity of these therapeutic agents. Indeed, the host–microbiota interface is a rich signaling hub that integrates environmental inputs with genetic background and immune features to shape the host immune system. Often, hematopoietic and non-hematopoietic cells reside strategically in this interface with the aim of sensing microorganisms and their byproducts or metabolites to translate them into the host physiological response [37].

Differential gut microbiome composition was found to influence the clinical outcome, the incidence, and the severity of immune-related colitis associated with ipilimumab treatment in metastatic melanoma patients [36]. This study unveiled the microbiota composition in relationship with the insurgence of irAEs and reported that patients whose gut microbiome was enriched with bacteria belonging to the Bacteroidetes phylum were less likely to experience colitis, while an enrichment with bacteria belonging to the Firmicutes phylum correlated with a higher probability of developing colitis [36]. These findings have also relevant clinical implications, as demonstrated by a study involving two patients with treatment-refractory ICI-related colitis, who had complete resolution of the clinical symptoms of colitis following fecal microbiota transplantation from a healthy donor [38]. The patients were deficient in Bacteroides at the point of colitis manifestation, while they were found to be enriched in Bifidobacterium following the treatment.

The variety of the microbiota composition is of crucial relevance, and this could be influenced by several factors, including but not limited to age, long-term diet habits, and racial/ethnic belonging. The contribution of the latter was investigated by Yoshimura and colleagues in a study involving 26 Japanese cancer patients, as the Japanese population is knowingly enriched in Bifidobacterium, while lacking Bacteroidetes and Prevotella compared to other populations [91]. They reported that responders to anti-PD1 treatment had higher proportions of Catenibacterium, Turicibacter, and species belonging to the Prevotella and Parabacteroides genera, while non-responders were particularly enriched in Desulfovibrion. Interestingly, when patients were stratified according to irAE occurence, it was demonstrated that the proportions of Turicibacter and Acidaminococcus were sensibly higher in the group with irAEs, while the proportions of Blautia and Clostridiales were higher in the group not experiencing toxicities. Additionally, responders experiencing irAEs were enriched with Acidaminococcus and Turicibacter, while responders not experiencing toxicities were enriched with Blautia and Bilophila [92].

Another factor influencing the gut microbiome composition is the concomitant use of antibiotics that compromise the intestinal microbial biodiversity. Indeed, a retrospective study evaluating the insurgence of intestinal irAEs in solid tumor patients treated with ICIs reported that 92% of patients exposed to proton–pump inhibitors developed gastrointestinal toxicities [93].

Altogether, these studies demonstrate how the gut microbiota could represent both a therapeutic opportunity and a predictive biomarker in the context of irAEs. Computational methods based on random forest algorithm can be exploited to develop classifiers based on the presence of specific microbial populations capable of discriminating in silico irAEs vs. non-irAEs cohorts, with the potential of predicting irAE onset [94].

### 4.2. Genetic Background

The host genetic background is thought to play a role in the development of irAEs, and potential genetic markers predispose individuals to the onset of irAEs.

A pilot exploratory study involving melanoma patients treated with ICIs and experiencing toxicities demonstrated that several autosomal genetic variants were associated with the occurrence of these toxicities. Precisely, they identified 30 single-nucleotide polymorphisms, among which 12 were associated with an increased risk of irAE development, mapping unique genes associated with autoimmunity or inflammatory diseases [39].

Interestingly, the similarity to autoimmune diseases suggests a possible link between the susceptibility to develop irAEs and the presence of SNPs/genetic alterations representing risk loci for autoimmune diseases. The most relevant susceptibility loci that could potentially predict treatment-induced irAEs were extensively reviewed by Hoefsmit and colleagues, who reported that multiple irAEs’ risk loci are shared with autoimmune diseases and are responsible for diverse immune functions, such as antigen presentation, cytokine signaling, NF-κβ transcriptional regulation, or T cell activation/inhibition [95]. For example, polymorphisms in genes encoding macrophage mediators and Fcγ receptors are associated with the onset and severity of neuropathies, while polymorphisms in MHC genes are thought to be genetic predisposing factors for the occurrence of myocarditis. Additionally, thanks to the “Immunochip project”, relevant loci have been identified in genes involved in the T-cell receptor signaling pathway as susceptibility factors for the development of thyroiditis and endocrinopathies [95]. Moreover, certain Human Leukocyte Antigens (HLA) types could be predisposed to specific organ-specific toxicity [40]. In fact, various groups reported that the majority of patients with ICI-related arthritis possess Rheumatoid arthritis (RA)-associated HLA-DR susceptibility alleles [77]. It is also frequent that patients with ICI-related diabetes have at least one HLA-DR risk allele, as HLA-DR4 predominance has been reported in patients with ICI-induced diabetes [33,74,96]. Additionally, HLA-DRB1*04:05 has been associated with ICI-induced arthritis and HLA-DRB1*11:01 with the development of pruritus or colitis during therapy [77,97].

Collectively, these findings indicate that autoimmune disease-associated susceptibility loci could represent potentially good candidate biomarkers to predict immune-mediated toxicities of ICIs.

### 4.3. Pre-Existing Autoimmunity

Concerns of irAE onset have directly impacted the consideration of autoimmune patients in cancer treatments. Clinical trials evaluating immune checkpoint inhibitors often exclude patients with diagnosed autoimmune diseases.

Only recently, reports of the administration of ICIs in patients with pre-existing autoimmunity have emerged, showing that these patients have higher rates and severity of irAEs. In fact, in a study involving 30 autoimmune patients treated with ipilimumab for advanced melanoma, it was reported that 27% of patients experienced exacerbations of the autoimmune condition and 33% developed high-grade irAEs [98].

Various studies have investigated the influence of ICI usage in autoimmune patients on the induction of toxicities, and they are extensively reviewed in the work by Ibis and al [99]. Psoriasis patients have the highest risk of developing flare or de novo irAEs after ICI treatment, and up to 60% of patients with RA experience flares of their disease. Patients with gastrointestinal and rheumatologic autoimmune diseases had the most frequent flare-ups after ICI therapy, whereas patients with Hashimoto thyroiditis and neurological autoimmunity developed mostly new irAEs [99].

This is often synonymous with efficient immune system activation, which could be beneficial for tumor treatment. Indeed, Gulati et al. showed that melanoma patients with pre-existing autoimmune disorders had increased progression-free survival and development of irAEs when treated with ICI. On the other hand, patients receiving ICIs with pre-existing autoimmunity who survived for more than 1 year developed new-onset chronic kidney disease [100].

Interestingly, disease flares were increased in patients receiving anti-PD/DL1 treatment, while de novo irAEs were observed more often in patients treated with anti-CTLA4. Additionally, a retrospective study showed that patients who received ICI and had a pre-existing autoimmune condition (most commonly being RA, psoriasis, and polymyalgia rheumatica) developed cardiovascular irAEs more often than patients without autoimmunity, indicating that patients with these pre-existing autoimmune conditions should be monitored for cardiac toxicities [101].

The higher risk of the insurgence of toxicities in autoimmune patients can be ascribed to the higher baseline status of inflammation, characterized by higher levels of autoAbs, cytokines, and immune cells. In fact, the presence of these circulating biomarkers has been proposed as a mechanistic link between autoimmunity and irAEs, as previously described in Section 3.3 and Section 3.5.

### 4.4. Patient-Specific Demographic Factors (Age, Sex, and Body Mass Index)

Evidence regarding the association between body mass index (BMI) and irAEs among cancer patients is scarce. Only recently, studies addressed the impact of body composition on the outcomes and toxicities of ICI treatments. The underlying mechanism is not well understood, even though obesity may promote leptin-mediated T-cell dysfunction, which can be reversed by blocking PD/DL1 [102]. Nonetheless, this relationship appears to be complex, and some studies failed to demonstrate an association [103,104].

On the other hand, a meta-analysis demonstrated that being overweight or obese was associated with increased odds of developing irAEs, and Young and colleagues reported diminished overall survival (OS) and progression-free survival among obese patients treated with ICIs for metastatic melanoma, compared to patients with a normal BMI [105,106]. In this study, the OS outcomes were gender-specific: significant associations were found only among males [106].

Ageing is linked to immune senescence and to alterations of the immune system that can lead to certain autoimmune diseases. Gender also seems to play a role, as the majority of autoimmune diseases are diagnosed in women [107]. Interestingly, the role of gender as a toxicity predictor was reported in a prospective study involving metastatic melanoma patients treated with ipilimumab; it was shown that female patients had an increased incidence of irAEs; thus, female gender was considered a significant and independent risk factor [108]. The effect of ageing was evaluated in a study involving 455 advanced melanoma patients treated with ICIs. The study reported that younger age was associated with more severe irAEs and hospitalizations, while older patients had more casualties and increased length of stay (LOS) when hospitalized [107].

## 5. Main Clinical Manifestations Due to CTLA4 and PD1 Pathway Inhibitors

The frequencies of the diverse clinical manifestations due to irAEs are differently distributed between CTLA4 and PD1 pathway inhibitors (Figure 3). The intricate interplay between patient-related factors in predisposing the occurrence of specific irAEs (e.g., immune system alterations, such as pre-existing autoimmune conditions), the differential expression of the antigens targeted by these monoclonal antibodies in different tissues (CTLA4 is more frequently expressed—among others—in the gastrointestinal tract, as compared to the PD1 ligand), and the level at which these two immune checkpoints of T cells act in modulating the immune response (i.e., CTLA4 early, primarily in lymph nodes while PD1 later, primarily in cancer tissues) may explain why different frequencies of different irAEs are observed with anti PD1/PD-L1 versus CTLA4 inhibitors. Another factor to be balanced is that PD1 pathway inhibitors have collectively more indications compared to CTLA4 inhibitors, and this could represent a bias in the interpretation of data if not carefully considered (Table 1) [4,5,6,7,8,9].

Additionally, the time of onset of irAEs following ICIs varies depending on the type of ICI used. Generally, with PD-1 inhibitors, irAEs tend to have a later onset, typically occurring weeks to months after initiation of treatment [4,5,6,7,8,9]. In contrast, irAEs associated with CTLA-4 inhibitors often have an earlier onset, typically manifesting within the first few weeks of treatment [4,5,6,7,8,9].

According to the results of a recent meta-analysis conducted on 35 RCTs including more than 16,000 patients, hypothyroidism, hyperthyroidism, and pneumonitis occurred more frequently in individuals using PD1/PD-L1 inhibitors, whereas colitis and hypophysitis were more common in those using CTLA4 inhibitors [109]. The combination of ICIs significantly elevated the incidence of colitis and hypothyroidism compared to the use of PD1/PD-L1 inhibitors or CTLA4 inhibitors alone.

### 5.1. Gastrointestinal irAEs

Gastrointestinal manifestations represent the most frequent irAE due to ICIs (Table 2). Colitis is the most common immune-related toxicity and manifests with diarrhea, abdominal pain, rectal bleeding, fever, nausea, and vomiting [110]. A meta-analysis showed an overall incidence of all-grade colitis of 13.6% in patients treated with a combination therapy of ipilimumab/nivolumab, 9.1% in patients receiving ipilimumab, and 1.3% with anti-PD1/PD-L1 monotherapy [111]. The use of non-steroidal anti-inflammatory drugs can be a relevant risk factor, while inflammatory bowel disease is associated with severe ICI-induced colitis [110,112]. An early clinical and instrumental (colonoscopy) diagnosis of colitis is essential to avoid complications such as dehydration, toxic megacolon, colonic perforation (seen in 1–6.6% of patients), and death [7].

Gastritis and esophagitis are rare immune-related toxicities caused by ICI therapy that need to be differentiated from gastritis and esophagitis induced by *H. pylori* infection and the use of non-steroidal anti-inflammatory drugs. They represent about 3% of upper gastrointestinal immune-related symptoms and are diagnosed by performing esophagogastroduodenoscopy [113].

Cholecystitis from ICIs is also a rare irAE, and the cardinal symptoms are pain in the right upper quadrant of the abdomen, vomiting, diarrhea, and fever [113].

Pancreatitis from ICIs may occur in 4% of patients, who present the typical symptoms of non-immune-related acute pancreatitis, including epigastric pain, nausea, vomiting, fever, diarrhea, and elevated lipase with or without imaging abnormalities; however, some patients may be asymptomatic with an incidental finding of an elevated serum lipase [113]. Duodenitis with abdominal pain and diarrhea is rare and may be hypothesized when colitis is ruled out [113].

Immune-related hepatitis may be diagnosed in up to 17% of patients receiving ICIs; a higher incidence (25%) is reported when ipilimumab and nivolumab are used in combination [113]. Most ICI-related hepatitis develops within 3 months after the start of the treatment and is characterized by an elevation of alanine and aspartate transaminase, although indices of cholestasis may rise. Patients are generally asymptomatic, but some of them can present non-specific symptoms such as fever and rash, or rarely develop jaundice and liver failure [113].

### 5.2. Endocrinologic irAEs

Endocrine irAEs (E-irAEs) are one of the most common complications of cancer immunotherapy, occurring more frequently in patients treated with a combined regimen than in those treated with monotherapy (Table 2) [114]. The clinical course is similar to that of idiopathic autoimmune endocrinopathies. In the first inflammatory phases, patients are usually asymptomatic, and diagnosis depends on abnormal laboratory findings. Once clinical manifestations appear, most hormone-producing cells have already been destroyed [115]. This explains the lack of response to steroid treatment and the irreversible nature of endocrine toxicity, a unique feature in the landscape of irAEs [116,117]. The treatment of E-irAEs is thus mainly based on life-long hormone supplementation.

The most frequent E-irAE is thyroid dysfunction. Incidence is different according to the ICI subtype, occurring in 5% of patients treated with anti-CTLA4 and in 10% of patients treated with anti-PD1/PD-L1 [114]. Thyroid dysfunction is the consequence of immune-mediated thyroiditis. Except for some patients presenting with symptomatic hyperthyroidism, the early phases are usually asymptomatic until hypothyroidism becomes clinically relevant, requiring hormone supplementation. Anti-thyroid Abs are detectable in 100% of cases, and several pieces of evidence exist regarding the higher risk of developing ICI-related thyroiditis in patients with Abs positivity prior to ICI administration [118,119]. Interestingly, the development of thyroid dysfunction has been demonstrated to associate with improved survival and better outcomes [120].

Autoimmune hypophysitis, usually considered a rare condition, is relatively frequent during ICI treatment, mostly induced by direct toxicity of anti-CTLA 4 or combined treatment, or alternatively by anti-thyrotrophs, corticotrophs, and gonadotrophs Abs [78,114]. The clinical picture depends on the amount of the endocrine axes affected by the process and consequently on specific hormone deficiencies. The spectrum of autoimmune hypophysitis comprises secondary adrenal insufficiency, central hypothyroidism, hypogonadism, and central diabetes insipidus, along with headache and visual loss, due to possible mass effect with consequent cerebral oedema [121,122].

Other less common E-irAEs are ICI-induced primary adrenal insufficiency (1–5%) and autoimmune diabetes mellitus (DM), both potentially life-threatening if not promptly recognized [123,124]. ICI-related DM results from autoimmune destruction of pancreatic β-cells, and most patients present with diabetes ketoacidosis. Unlike type 1 DM, autoAbs are detectable only in half of patients, and possible exocrine dysfunction may occur [74,124].

### 5.3. Cutaneous irAEs

Immune-related cutaneous toxicity may develop in 30–60% of patients receiving ICIs (Table 2), and it is more frequent with anti-CTLA4 monotherapy (44–59%) than with anti-PD1 (34–42%) and anti-PD-L1 (20%). It increases when anti-CTLA4 and anti-PD1 are combined (59–72%). Also, severe irAEs (Grade 3–4, according to Common Terminology Criteria for Adverse Events (CTCAE)) are higher with combination therapy (14.5%) than with monotherapy with anti-CTLA4, PD1, and PD-L1 (4.7%, 7.2%, 2.3%, respectively) [125,126,127,128]. The clinical manifestations of irAEs are very heterogeneous and differ according to the type of cancer treated with ICIs.

The maculo–papular rash is the most frequent irAE and occurs quite early during ICI treatment. It presents as erythematous macules and papules and rash, which may affect every part of the body (face, scalp, trunk, and extremities) [7,128]. Xerosis is often associated with eczematous dermatitis, particularly in patients treated with anti-PD1. It consists of papules and macules usually on the trunk and extremities, although the face may also be affected by seborrheic dermatitis [128].

Psoriasis may be divided into two groups: de novo psoriasis (new onset) and reactivated psoriasis (worsening or recurrence of pre-existing psoriasis). The typical manifestation appears as scaly erythematous plaques with well-defined borders on the trunk and extremities, while in some cases, palms/soles are involved, and small-sized rashes present as guttate-type psoriasis. Patients with a history of psoriasis develop cutaneous toxicity earlier after starting ICIs than those with de novo psoriasis [127,128].

Lichen planus/Lichen planus-like eruptions may develop in up to 6% of patients treated with ICIs, more frequently with anti-PD/PD-L1 than anti-CTLA4. They are characterized by rashes associated with itchy, red to violaceous, flat-topped papules or plaques on the extremities and trunk, while they rarely involve the oral mucosa, unlike spontaneous Lichen planus [127,128].

Bullous diseases, such as bullous pemphigoid (BP) and mucous membrane pemphigoid, are uncommon cutaneous irAEs from ICIs; they can occur anywhere from 3 weeks to 20 months after the initiation of immunotherapy. BP usually manifests with tense bullae accompanied by or preceded by pruritus, but in some cases, it consists only of urticarial and eczematous plaques with pruritus [127,128].

Vitiligo is a cutaneous toxicity resulting from a loss of melanocyte function in the epidermis. It is more frequently observed in patients with melanoma treated with anti-CTLA4 drugs (2–9%), anti-PD-L1 (7–11%), or combinations, while it is less frequent in patients treated with ICIs for other tumors. ICI-induced vitiligo affects photo-exposed areas, with flecked macules that coalesce into patches [128].

Alopecia may develop in about 1–2% of patients receiving ICIs, with a phenotype similar to that of alopecia areata and a clinical condition that may vary from well-defined patches or diffuse hair loss on the scalp to alopecia universalis [128].

Pruritus is the second most common cutaneous irAE associated with ICIs, with an incidence of 13–20%. Symptoms are commonly mild: most patients develop G1 or G2 events, while severe grade events are more common with anti-CTLA4 and combination treatments. Pruritus may involve every body part, particularly the trunk and scalp, whereas the face, soles, anterior neck, and genitals are rarely involved [127,128]. Moreover, it is often associated with other skin manifestations such as erosions, ulcerations, hyperpigmentation, or prurigo nodules.

Stevens–Johnson syndrome and toxic epidermal necrolysis are rare but life-threatening cutaneous irAEs. Clinical manifestations include high fever, widespread detachment of the epidermis, erosions, and mucositis. Drug-induced hypersensitivity syndrome (DIHS)/drug reaction with eosinophilia and systemic symptoms (DRESS) are other rare irAEs from ICIs; they may cause renal and hepatic impairment [127,128].

### 5.4. Pulmonary irAEs

Immune-related pulmonary toxicities are less common than others, but they can be fatal or cause therapy withdrawal (Table 2). The most common pulmonary irAE is pneumonitis or interstitial lung disease (ILD), which is a localized or diffuse inflammation of the lung parenchyma that may develop at any time during ICI treatment, with early-onset forms having a more severe course [7]. According to a meta-analysis, the incidence of any grade and severe (grade ≥ 3) pneumonitis is 2.7% and 0.8%, respectively [129]. Moreover, the incidence depends on the tumor type and ICI used: specifically, it is higher in lung cancer and with PD-L1 inhibitors or combination therapy (including nivolumab and ipilimumab). On the contrary, a combination of pembrolizumab with chemotherapeutic agents was not related to an increased risk of ICI-related ILD in patients with lung cancer, and the administration of durvalumab after chemoradiation in patients with locally advanced NSCLC was also associated with an acceptable rate of pulmonary toxicity [129,130]. Risk factors of ILD are older age, tobacco smoke, history of lung disease, previous lung surgery, or radiotherapy. The main clinical manifestation is breathlessness and/or other respiratory symptoms, with inflammatory findings on chest CT following treatment with ICI [130].

Other immune-related pulmonary toxicities have been reported in the literature, but they are very rare, and most information comes from case reports: sarcoidosis, airway disease (mainly bronchiolitis and asthma), pulmonary vasculitis, lung infections, pulmonary nodules, diaphragm myositis, allergic bronchopulmonary aspergillosis [130].

### 5.5. Musculoskeletal irAEs

Rheumatic irAEs are reported in a range of 5% to 10% of patients receiving ICIs, although in real-life reports, the incidence rises up to 22% (Table 2) [131]. However, the precise incidence may differ for individual rheumatic manifestations, such as inflammatory arthritis, polymyalgia rheumatica, myositis, large vessel vasculitis, and sicca syndrome [132].

A prevalent manifestation of rheumatic irAEs is inflammatory arthritis, exhibiting different clinical phenotypes, ranging from oligoarticular to polyarticular involvement, but more often presenting symmetrically and involving both large and small joints (RA-like phenotype) [133]. The onset of arthritis displays variability, occurring either early in the course of ICIs or several months thereafter. Clinical phenotypes of inflammatory arthritis may be accompanied by the presence of autoAbs such as RF and anti-cyclic citrullinated peptide (anti-CCP), or it could be linked to familial history of psoriasis or inflammatory bowel disease [134]. Understanding the distinct clinical phenotypes of inflammatory arthritis, specific autoAbs involved, and identifying predisposing factors is paramount for delineating the specific rheumatic irAE. A comprehensive approach that integrates clinical, serological, and immunological parameters is necessary for accurate diagnosis and optimal patient care. Frequencies of arthritis as an irAE hinge on the specific checkpoint inhibitor employed, with PD1/PD-L1 inhibitors generally associated with higher incidence rates compared to CTLA4 inhibitors [131].

Polymyalgia rheumatica, characterized by pain and stiffness primarily affecting the shoulders and hips, is more frequently reported with PD1/PD-L1 inhibitors. Its occurrence and time of onset during immunotherapy exhibit variability [135]. Furthermore, vasculitis, characterized by inflammation in the blood vessels, and specifically, giant cell arteritis/large vessel vasculitis, represents a distinct irAE whose onset and frequency differ based on the specific ICIs utilized, being more frequently reported with anti-PD1/PD-L1 inhibitors, with the highest incidence when the combination of anti-PD1/PD-L1 and CTLA4 inhibitors are used [131].

Sicca syndrome, marked by dryness of mucous membranes, including eyes and mouth, is yet another documented rheumatic irAE. The development of sicca syndrome can manifest at different intervals during immunotherapy, being more frequently reported with anti-PD1/PD-L1 inhibitors [136].

Lupus-like disease, sarcoidosis, and systemic sclerosis are rarely reported, and they are generally induced by anti-PD1/PD-L1 inhibitors [7,127,128,132].

The prevalence of rheumatic irAEs is influenced by several factors, including the type of cancer being treated and the specific ICI employed. Patients with melanoma, lung cancer, and renal cell carcinoma may exhibit varying susceptibilities to these adverse reactions [137]. Notably, patients with a history of rheumatic disorders, such as RA or lupus, may have an increased predisposition to developing rheumatic irAEs during cancer immunotherapy; up to 71%, according to recent studies [138]. Rheumatic irAEs are not substantially higher in randomized controlled trials on ICIs compared to conventional treatment, indicating a potential discrepancy with real-life incidence and suggesting the likelihood of underreporting or misclassification [32].

### 5.6. Hematological irAEs

The real spectrum of hematological irAEs is not easy to characterize, since most studies do not discriminate between immune-mediated adverse events and other non-specific hematological toxicities (Table 2). The pathogenesis of most of these irAEs still remains to be elucidated. The response to glucocorticoid treatment supports the hypothesis of an immune-mediated process occurring centrally or peripherally [139,140]. Also, for those primarily involving the bone marrow, some authors suggest activation of T cells leading to dysregulation of T-helper cell cytokine secretion and T-cytotoxic cell tissue infiltration, with consequent dysfunction in hematopoietic cell maturation and proliferation processes [141]. Their frequency is low compared to other irAEs and has been estimated at 3.6% for all grades and 0.7% for grades III–IV [139]. It seems that an overall higher frequency of hematological irAEs occurs with anti-PD1/PD-L1 compared to anti-CTLA4 [139].

The most commonly reported hematological irAE is autoimmune hemolytic anemia (AIHA), accounting for more than 25% of cases [139,140,142]. The antiglobulin (Direct Coombs) test is usually positive for complement 3d or for IgG. Most patients present with warm AIHA, although forms with cold Abs or cold agglutinins have also been reported [140,143].

Immune thrombocytopenia usually presents with a severe (grade IV) isolated abnormal platelet count in most of the patients. Nevertheless, hemorrhagic symptoms are reported in only 20% of cases [140]. Serum antinuclear Abs (ANA) may be detected, but most of the patients do not present other Abs positivity [140,144].

Immune-mediated neutropenia is usually severe and life-threatening due to the risk of consequent bacterial and fungal infections. In these cases, treatment with granulocyte colony-stimulating factor (G-CSF) alone may be not effective, and the efficacy of steroid administration supports the immune pathogenesis of this condition [140]. A high risk of relapse has been reported in cases of ICI rechallenge despite complete resolution of the first event [140].

Hemophagocytic lymphohistiocytosis (HLH) is less common and, in contrast to other hematological irAEs, is more frequently linked to anti-CTLA4 treatment [139,142,145]. HLH also differs from other hematological toxicities due to the high mortality rate despite immunosuppressive treatment [139]. Interestingly, one study reported a concomitant Epstein–Barr virus infection in 20% of patients, suggesting the need of another trigger such as a chronic underlying infection [142].

### 5.7. Cardiovascular irAEs

Even though it is difficult to define the real incidence of cardiovascular irAEs, most of the data come from case reports, retrospective case series, and nationwide cohort studies (Table 2) [146,147].

Myocarditis represents the most common cardiovascular adverse event due to ICIs. It can have an early onset after the starting of ICI, generally within 3 months, and can mainly affect male patients aged 70–79 years with melanoma, NSCLC, or renal cancer [146]. A higher incidence is associated with the nivolumab plus ipilimumab combination (0.27%) vs. nivolumab alone (0.06%); unfortunately, 50% of ICI-induced myocarditis cases are fatal [146].

The symptoms of myocarditis are shortness of breath, tachycardia or palpitations, fatigue, chest pain, cough, and episodes of syncope. Specific ocular and neuromuscular symptoms may suggest concomitant myositis and/or myasthenia gravis, configuring an overlap syndrome, diagnosed in up to 25% of cases [148].

Pericardial disease is the second most common ICI-associated cardiotoxicity, and it is associated with higher mortality rates (21%) [148,149]. The main symptom is shortness of breath, but precordial pain, jugular venous congestion, or cardiogenic shock (in the case of cardiac tamponade) have also been reported [149].

Tachyarrhythmias (atrial fibrillation and sinus tachycardia) and bradyarrhythmias (atrioventricular block) may develop within the first six months of ICI treatment. They are observed mostly in male older patients with a history of hypertension, diabetes mellitus, and hyperlipidemia. Although some patients remain asymptomatic, most of them develop palpitations, fatigue, dyspnea, dizziness, or syncope [148].

Takotsubo cardiomyopathy, acute heart failure, and acute coronary syndrome are the rarest ICI-induced cardiac toxicities. They usually are associated with chest pain and shortness of breath and require an early diagnosis, both clinically and instrumentally (electrocardiogram, cardiac ultrasound, and laboratory tests) [7,146,148,149].

### 5.8. Nephrological irAEs

Immune-related renal toxicity is a relatively infrequent event during ICI treatment; its incidence has been reported in up to 6% of patients (Table 2) [7,150,151].

The most common event is represented by acute kidney injury (AKI), defined as an increase in serum creatine of at least 1.5 x the upper limit of normal. The clinical diagnosis, which has to be confirmed by a renal biopsy, must include at least two of the following: increased sCr ≥ 50% or the need for renal replacement therapy (RRT) and sterile pyuria (500 white blood cells/hpf) or eosinophilia (500 cells per L) [152]. The most typical histopathological finding of AKI is acute tubular interstitial nephritis, but less frequent histological lesions (pauci-immune vasculitis or thrombotic microangiopathy) may be found. Beyond increased creatinine levels and pyuria, haematuria and proteinuria can lead to suspicion of AKI. ICI-related AKI may develop from 3 to 12 months from the start of treatment, and it is more common with anti-PD-L1 ICI combined with either anti-CTLA4 ICI or chemotherapy than with ICI as a single agent.

Other renal irAEs are represented by acid-base and electrolyte disorders. The most common is hyponatremia, generally secondary to ICI-related endocrinopathies or renal tubular damage. Hypokalemia and metabolic acidosis may indicate distal tubular damage [150,153].

### 5.9. Neurological irAEs

Neurological irAEs (N-irAEs) are rare compared to other irAEs and occur in 1–5% of patients treated with ICI [154,155,156]. Nevertheless, they are frequently severe, with about 1% of patients presenting with Grade III or IV adverse events, often requiring ICI discontinuation along with immunosuppression [154,156]. They usually occur within 4–6 months after the first treatment infusion, but delayed presentations have been also described [157,158]. The spectrum of N-irAE includes a wide variety of clinical phenotypes, with possible involvement of the central nervous system (CNS), peripheral nervous system (PNS), or both (Table 2).

Neuromuscular disorders are the most common clinical presentation, accounting for 75% of all N-irAEs, whereas CNS involvement occurs in 25% of patients [159]. The clinical phenotype is partially influenced by the ICI subtype. Meningitis has been reported to be more frequent in patients treated with anti-CTLA4 and less common in anti-PD1/PD-L1, whereas myasthenia gravis and myositis are more often associated with anti-PD1/PD-L1 [159]. The use of ICI-combined regimens usually increases the risk of developing N-irAEs, with some phenotypes like cranial neuropathies occurring more frequently in patients exposed to both ICIs [156,159]. Clinical presentation has been also demonstrated to depend on the underlying cancer types, and some clinical associations have been recently described. For instance, a diagnosis of melanoma has been more frequently reported in patients with polyradiculoneuropathy and meningitis and less commonly in patients with encephalitis, more often described in association with lung cancer [158,159]. Some of these associations may be explained by distinct pathogenic mechanisms underlying different N-irAEs. Similarly to other non-neurological irAEs, some complications derive from a silent pre-existing and cancer-independent autoimmune condition, which may result in clinically manifest disease only after ICI-mediated immune activation in predisposed subjects [160]. By contrast, for other N-irAE, ICIs act as a second hit, boosting the immune response against antigens shared between cancer and neural tissue, as in classical paraneoplastic syndromes (PNS) [160,161,162]. Similarly to non ICI-related PNS, these “PNS-like” N-irAEs more frequently occur in association with lung cancer, explaining the greatest incidence of these phenotypes in the context of this tumor. These N-irAEs resemble PNS both in terms of clinical presentation and outcome, with greater disease severity that often requires long-term immunosuppression and poorer prognosis [158,159]. Regardless of the different pathogenesis, N-irAEs usually present with specific features different from conventional non-ICI-related forms (described below).

Myositis is the most frequent N-irAE [157,158,159]. Clinical presentation ranges from oligosymptomatic hyperckemia to severe forms [159,163]. It usually presents with limb-girdle weakness, often associated with bulbar, ocular and facial involvement, which are more rare in idiopathic forms [163,164]. Respiratory dysfunction and myocarditis occur in 30% of patients [163,164]. Abs anti AchR are detectable in 40–60% of patients independently of clinical signs of myasthenia gravis [158,165]. Except for rare forms of anti TIF-1γ positive dermatomyositis, myositis-specific Abs are usually absent [159,163]. Muscle biopsy differs from other inflammatory myopathies, presenting with a specific pattern of focal clusters of necrosis with possible endomysial infiltrate of T cells [163,164]. ICI-related myositis is usually steroid-responsive, but mortality due to cardiac and/or respiratory involvement is observed in 15% of cases [158,159,164].

ICI-related Myasthenia Gravis usually differs from classical forms because of frequent bulbar and respiratory involvement, which occur in more than half of patients [159,166,167]. By contrast, isolated ocular forms are uncommon. As already described for myositis, there is a clear overlap between myasthenia and muscular involvement, with 50% of patients presenting with concomitant myositis with or without myocarditis [159,166,168]. Anti-AchR Abs are detectable in most patients [159,166]. Despite a favorable outcome in most of the patients, a higher mortality rate has been reported for ICI-related myasthenia gravis compared to classical forms because of the high frequency of respiratory failure and cardiac involvement [157,158,166].

ICI-related neuropathies are a wide, heterogeneous group. The most frequent phenotype is acute/subacute demyelinating polyradiculonevritis, which usually presents with predominant motor involvement and concomitant affection of cranial nerves [159,169]. Unlike classical Guillan Barrè, pleocytosis is often detected in cerebrospinal fluid, and a rapid response to glucocorticoids is usually observed. The outcome depends on respiratory involvement, and ICI withdrawal is typically necessary due to the high risk of relapses [158,169]. Another important presentation is sensory neuronopathy. It is clinically indistinguishable from its classical paraneoplastic variant, with asymmetric onset and predominant involvement of the upper limbs, poorly responsive to treatment, and often associated with anti-onconeural Abs [158,159]. Lastly, cranial neuropathy more frequently affects patients treated with a combination of different ICIs [159]. All cranial nerves may be affected, but it presents more often with facial palsy or painless neuritis [169,170].

Focal encephalitis is the most frequent CNS N-irAE, occurring in more than 50% of cases [159,171]. Several phenotypes have been described, but most patients present a well-defined limbic or brainstem encephalitis, whereas isolated cerebellar syndromes are less common [159,172]. Clinical presentation and imaging findings resemble those of classical PNS [159,172]. Onconeural Abs are detectable in more than half of patients, with the most reported specificity being anti Ma2-Abs, unlike in classical PNS, where anti-Hu-associated encephalitis is more frequent [159,173]. Response to glucocorticoids is mostly unsatisfactory, and intensive immunosuppression along with permanent ICI discontinuation is often required [158].

Meningitis and meningoencephalitis are less common and usually present with fever, meningeal signs with or without encephalopathy [171,172]. In some cases, anti-GFAP Abs may be detected [172]. Other atypical CNS phenotypes are myelitis and other demyelinating syndromes. In some of these patients presenting with well-defined Neuromyelitis Optica, anti-aquaporin 4 Abs have been reported [159].

## 6. Current Limitations and Future Perspectives

IrAEs represent a unique challenge in modern medicine, and our understanding of them is evolving rapidly. There is growing consensus regarding their pathophysiology and management, but a gap remains between these aspects and clinical practice, where mechanisms are not entirely clear or where conflicting evidence exists. In particular, prompt identification and treatment still represent a clinical challenge, considering the heterogeneity of presentation of irAEs [6,7,8]. In addition, the correlation between irAEs and treatment outcomes, highly specific biomarkers, how to tailor immunosuppression on ICI outcomes, and the management of steroid-refractory cases are clinical challenges [1,2,3,4,5,6,7,8,9]. Probably, more sophisticated preclinical models of irAEs, linking oncology to systemic and organ autoimmunity, will be developed, and clinical studies carrying important information on risk stratification, biomarkers, and more evidence-based approaches for the management of irAEs will be conducted. These advances will support optimal care for all patients receiving ICIs, guided by cross-pollination of the expertise and insights gained from multidisciplinary teams.

## 7. Conclusions

In recent years, the treatment algorithm of almost all cancer types has been completely revolutionized by the introduction of ICIs, alone or in combination with chemotherapy. Patients receiving these drugs often report long-lasting clinical and radiological responses, but they are at risk of developing irAEs. Most of the toxicities are mild and easily manageable, while less than 10% of them may be severe and need to be recognized and promptly treated.

In the present work, we reviewed the molecular mechanisms of ICIs and their role in the regulation of the immune system, boosting the antitumor immune response. We analyzed the complex pathogenic mechanisms responsible for the development of irAEs, which result from the evasion of self-tolerance checkpoints consequent to drug-related immune modulation. Furthermore, we reviewed the host-specific and environmental factors predisposing to irAEs and the clinical manifestations by organ types. We reported that irAEs occur more often with ICI combinations, with neurological, cardiovascular, hematological, and rheumatic irAEs more frequently associated with PD1/PD-L1 inhibitors, while cutaneous and gastrointestinal irAEs are more frequently associated with CTLA4 inhibitors.

Due to their nonspecific signs and symptoms, some irAEs may be difficult to diagnose, especially those occurring less frequently. The differential diagnosis with autoimmune disorders may be challenging, given the pathophysiology and clinical similarities.

Considering the increasing use of ICIs, this area of research necessitates further clinical and preclinical research, bridging the gap between basic science and clinical practice, as well as greater education among clinicians and patients.

## Figures and Tables

**Figure 1 cancers-16-01440-f001:**
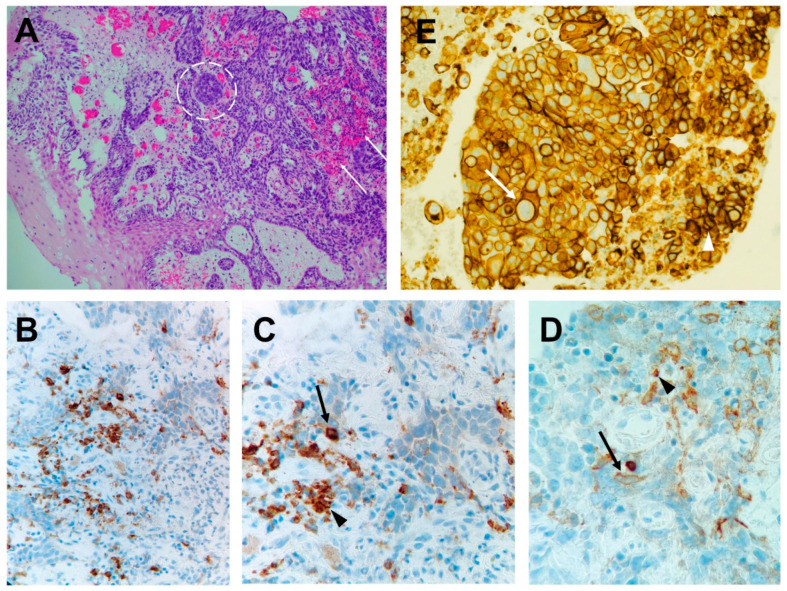
**Programmed cell death protein ligand 1 (PD-L1) expression in tumor cells and immune infiltrate.** (**A**) Esophagus squamous cell carcinoma 2.5 × Hematoxylin-eosin stain: neoplastic cells (circled), inflammatory infiltrate (arrows). (**B**) Esophagus squamous cell carcinoma 20× stained with monoclonal antibody anti-PD-L1. (**C**,**D**) Esophagus squamous cell carcinoma 40× stained with monoclonal antibody anti-PD-L1. Arrow: membrane positivity of neoplastic cell. Arrowhead: membrane positivity of immune infiltrate. (**E**) Lung adenocarcinoma 40× stained with monoclonal antibody anti-PD-L1. Arrow: membrane positivity of neoplastic cell. Arrowhead: membrane positivity of immune infiltrate.

**Figure 2 cancers-16-01440-f002:**
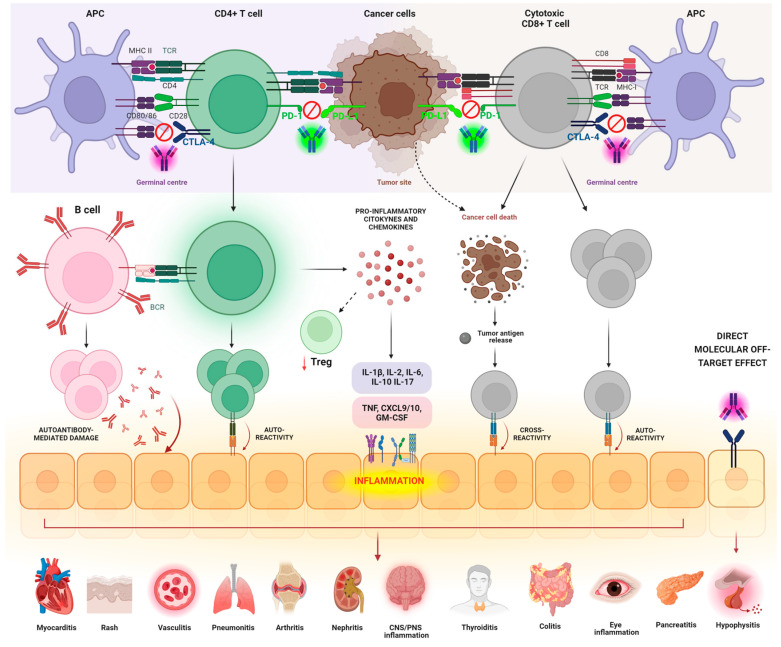
**Immunological mediators and molecular mechanisms driving ICI-induced irAEs.** Loss of immune checkpoint inhibition induced by ICI usage results in the diversification of T- and B-lymphocyte populations, with the consequent expansion of autoreactive clones that mediate direct damage to the tissues via production of inflammatory cytokines/chemokines and autoAbs. Alternatively, tumor cells release self-antigens following cytotoxic attack, which lead to cross reactivity with antigens found in normal tissues. Additionally, ICIs can mediate tissue damage via direct off-target binding to tissues expressing CTLA4. APC, Antigen Presenting Cell; MHC II, Major Histocompatibility Complex class II; TCR, T-cell receptor; CD4, Cluster of differentiation 4; CTLA4, Cytotoxic T-Lymphocyte Antigen 4; PD1, Programmed cell death 1; PD-L1, Programmed cell death ligand 1, CD8, Cluster of differentiation 8; MHC I, Major Histocompatibility complex class I; BCR, B-cell receptor; IL, interleukin; TNF, tumor necrosis factor; GM-CSF, Granulocyte–Macrophage Colony-Stimulating Factor. Created with BioRender.com.

**Figure 3 cancers-16-01440-f003:**
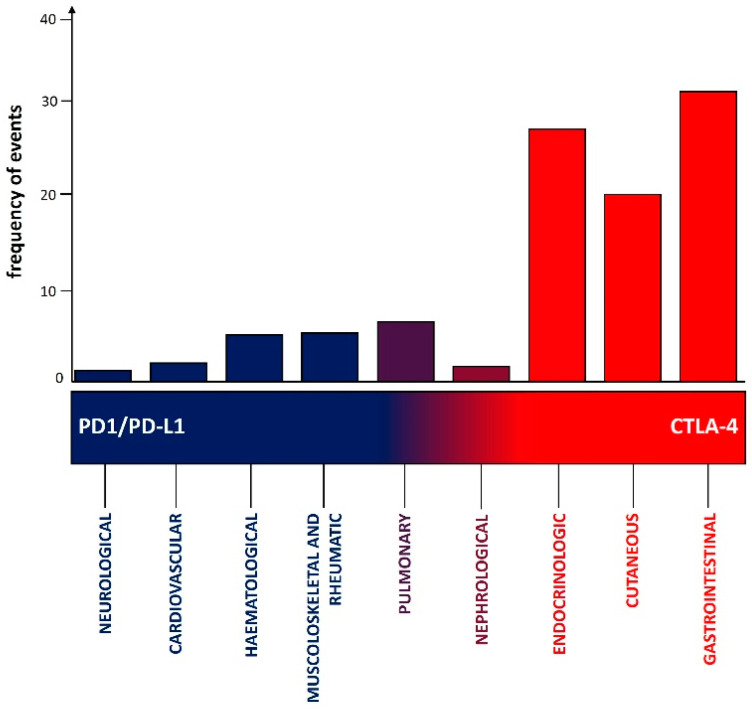
**Risk profiles of immune-related adverse events (irAEs) by different immune-checkpoint inhibitors (ICIs).** Representative irAEs from two systematic reviews are reported [30,109]. These irAEs have been grouped per type of organ involved. The frequency of occurrence is coherent with the length of the bars. The colors are indicative of the major involvement of a specific ICI: red indicate that the immune toxicity is more commonly associated with the use of anti-CTLA-4; blue indicate that the toxicity is more commonly associated with the use of PD1/PD-L1 inhibitors. Every category is the sum of all the system features, e.g., neurological irAE includes myositis (very frequent event, more often PD1-mediated) and meningitis (very rare event, more often CTLA4-mediated); therefore, the localization of the neurological irAE closer to PD1/PD-L1 instead of CTLA4 is driven by the higher frequency of myositis in this category.

**Table 1 cancers-16-01440-t001:** Clinical applications of approved immune checkpoint inhibitors.

Checkpoint Targets	Drugs	Indications
CTLA-4	Ipilimumab ^1, 2, 3, 5^	NSCLC, advanced CRC, advanced MM, HL
	Tremelimumab ^1, 2^	NSCLC, HCC
PD-1	Nivolumab ^1, 2, 3, 4, 5^	HNSCC, NSCLC, GC, ESGC, CRC, HCC, RCC, HL, MM, skin cancer
	Pembrolizumab ^1, 2, 3, 4, 5^	HNSCC, NSCLC, GC, ESGC, CRC, HCC, RCC, HL, MM, skin cancer, TNBC
	Cemiplimab ^1, 2, 3^	skin cancer, NSCLC
	Sintilimab ^1, 2, 5^	NSCLC, HCC, HL
	Camrelizumab ^2, 5^	NSCLC, HCC, HL, ESCC, HNSCC
	Dostarlimab ^1, 2, 3^	UCEC
	Tislelizumab ^1, 5^	NSCLC, HL, BRCA
	Penpulimab ^5^	HCC, GC, NSCLC, NPC, HL
	Toripalimab ^1, 5^	Skin cancer, HNSCC, BRCA
	Zimberelimab ^5^	HL
	Serplulimab ^5^	GC, CRC, NSCLC
	Pucotenlimab ^5^	CRC, MM
PD-L1	Durvalumab ^1, 2, 3, 5^	NSCLC, SCLC, BRCA
	Atezolizumab ^1, 2, 3, 5^	NSCLC, SCLC, BRCA, skin cancer, HCC, TNBC
	Avelumab ^1, 2, 3^	skin cancer, RCC, BRCA
	Envafolimab ^5^	CRC
	Sugevalimab ^5^	NSCLC

^1^ EMA, European Medicine Agency; ^2^ FDA, United States Food and Drug Administration; ^3^ HC, Health Canada; ^4^ MHLW, Ministry of Health, Labour, and Welfare of Japan; ^5^ NMPA, National Medical Products Administration of China. BCC, basal cell carcinoma; BRCA, bladder urothelial carcinoma; CC, cervical cancer; CRC, colorectal cancer; ESCC, esophageal squamous cell carcinoma; GC, gastric cancer; HCC, hepatocellular carcinoma; HL, Hodgkin’s lymphoma; HNSCC, head and neck squamous cell carcinoma; MCC, Merkel cell carcinoma; MM, malignant melanoma; NSCLC, non-small cell lung cancer; NPC, nasopharyngeal carcinoma; RCC, renal cell carcinoma; SCLC, small cell lung cancer; TNBC, triple-negative breast cancer; UCEC, uterine corpus endometrial carcinoma.

**Table 2 cancers-16-01440-t002:** Presentations of most frequent irAEs due to ICIs by organs.

irToxicity	irAE	Clinical Presentation
**Gastrointestinal**	Colitis	Diarrhea, abdominal pain, rectal bleeding, fever, nausea, and vomiting
	Gastritis and esophagitis	Dyspepsia, epigastric pain, heartburn, reflux, nausea
	Cholecystitis	Abdominal pain at right upper quadrant, vomiting, diarrhea, and fever
	Pancreatitis	Epigastric pain, nausea, vomiting, fever, and diarrhea
	Duodenitis	Abdominal pain and diarrhea
	Hepatitis	Asymptomatic vs. mild (fever and rash) or severe (jaundice and liver failure) symptoms
**Endocrinologic**	Thyroiditis	Usually asymptomatic in the first phases (tachycardia due to possible hyperthyroidism) followed by signs and symptoms of hypothyroidism. Anti-thyroid Abs always positive
	Hypophysitis	Various combination of symptoms related to adrenal insufficiency, central hypothyroidism, hypogonadism, and/or central diabetes insipidus, along with headache and visual loss, due to possible mass effect with consequent cerebral oedema
**Cutaneous**	Maculopapular rash	Faint erythematous macules and papules that coalesce into plaques; rashes in the trunk and extremities
	Xerosis and eczematous conditions	Itchy, poorly demarcated, and erythematous macules and papules that coalesce into plaques and patches on the trunk and extremities; seborrheic dermatitis-like lesions on the face
	Psoriasis/psoriasiform eruption	Scaly erythematous plaques with well-defined borders on the trunk and extremities
	Lichen planus/lichen planus-like eruption	Rashes associated with itchy, red to violaceous, flat-topped papules, or plaques on the extremities and trunk, rarely on the oral mucosa
	Bullous diseases	Tense bullae accompanied by or preceded by pruritus; in some cases, only urticarial and eczematous plaques with pruritus
	Vitiligo	Flecked macules that coalesce into patches
	Alopecia	Well-defined patches or diffuse hair loss on the scalp, alopecia universalis
	Pruritus	May involve every body part, particularly trunk and scalp
	Stevens-Johnson syndrome and toxic epidermal necrolysis	High fever, widespread detachment of the epidermis, erosions, and mucositis
	DIHS/DRESS	Renal and hepatic impairment
**Pulmonary**	Pneumonitis or ILD	Breathlessness and/or other respiratory symptoms (fever, cough, chest pain)
	Sarcoidosis	Dry cough, dyspnea, fever, chronic fatigue, weight loss, chest pain
**Musculoskeletal and rheumatic**	Inflammatory arthritis	Joint inflammation leading to articular pain, swelling, warmth, and stiffness (particularly if morning stiffness lasts more than 30 min). Laboratory tests, such as C-reactive protein and erythrocyte sedimentation rate, are often elevated. It could exhibit different clinical phenotypes, ranging from oligoarticular to polyarticular involvement but more often presenting symmetrically and involving both large and small joints (rheumatoid arthritis-like phenotype)
	Polymyalgia rheumatica	Pain and stiffness primarily affecting the shoulders and/or hips, due to inflammation of periarticular structures
	Vasculitis	Inflammations of blood vessels; particularly but not limited to large-size blood vessels (e.g., giant cell arteritis)
	Sicca Syndrome	Marked by dryness of mucous membranes, particularly but not limited to the eyes and mouth.
**Hematological**	autoimmune hemolytic anemia	Severe anemia with antiglobulin (Direct Coombs) test is usually positive for complement 3d or for IgG
	Immune thrombocytopenia	Mostly asymptomatic severe abnormal platelet counts with possible haemorrhagic manifestations
	Immune-mediated neutropenia	Usually severe with bacterial and fungal infections
**Cardiovascular**	Myocarditis	Shortness of breath, tachycardia or palpitations, fatigue, chest pain, cough, and episodes of syncope. Possible overlap with myocarditis, myositis, and/or myasthenia gravis
	Pericarditis	Shortness of breath but also precordial pain, jugular venous congestion, or cardiogenic shock
	Arrythmias	Palpitations, fatigue, dyspnea, dizziness, or syncope,
**Nephrological**	Acute kidney injury	increased creatinine levels, need for renal replacement therapy, sterile pyuria (500 white blood cells/hpf), eosinophilia (500 cells per L), haematuria, proteinuria
	Electrolyte disorders	Hyponatraemia, hypokalaemia, metabolic acidosis
**Neurological**	Myositis	Limb-girdle weakness with frequent bulbar, ocular, and facial involvement. Respiratory dysfunction and myocarditis in 30% of patients. Possible overlap with Myasthenia
	Myasthenia Gravis	Muscle weakness and fatigue with daily fluctuations, frequent bulbar and respiratory involvement, presence of anti-AchR Abs. Possible overlap with myositis and myocarditis
	Acute/subacute demyelinating polyradiculonevritis	Predominant motor involvement and concomitant affection of cranial nerves. Frequent pleocytosis in cerebrospinal fluid
	Sensory neuronopathy	Sensory deficits with asymmetric onset, predominant involvement of upper limbs, often associated with anti-onconeural Abs
	Cranial neuropathy	All cranial nerves may be affected. More often, facial palsy or painless neuritis
	Focal encephalitis	Limbic encephalitis or Brainstem Encephalitis (cerebellar syndromes are less common). Onconeural Abs are often detectable (++ anti-Ma2-Abs or anti-Hu Abs)
	Meningitis and meningoencephalitis	Fever, meningeal signs with or without encephalopathy.

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
