# Peer review of "Immune-Related Adverse Events Due to Cancer Immunotherapy: Immune Mechanisms and Clinical Manifestations"

_cancers, 2024, doi:10.3390/cancers16071440_

Round 1

Reviewer 1 Report

Comments and Suggestions for Authors

This is a review paper titled "Immune-Related Adverse Events due to Cancer Immunotherapy: Immune Mechanisms and Clinical Manifestations".

This review thoroughly analyzes the immunological processes and clinical manifestations of immune-related adverse events (irAEs) linked to cancer immunotherapy with immune checkpoint inhibitors (ICIs). The authors provide a detailed explanation of how immune checkpoints, such as CTLA-4 and the PD-1/PD-L1 axis, play a crucial role in regulating the immune system. The review outlines the different clinical symptoms of irAEs in various organ systems, covers their frequency and risk factors, and highlights the importance of developing better biomarkers and educating clinicians and patients to improve the management of these side effects as the use of ICIs grows in cancer treatment.

authors are requested to

Explore the complex interplay between ICI-induced immune modulation and the triggering or exacerbation of autoimmune conditions.

Expand the discussion on genetic, and add section on environmental, or patient-specific factors that may influence the risk and severity of irAEs.

Comments on the Quality of English Language

Overall, the paper is well-written, but there is minor English language errors or typos that can be corrected.

Abstract, Line 26 : Change "Here we reviewed and discuss discussed the mechanisms….” To "Here we reviewed and discussed the mechanisms of action…”

P4L143-144:correct the grammatical erros and rewrite as follow “The expression of PD1 is a consequence of T and B cell activation and is mediated by the action of inflammatory cytokines, such as IFNγ, in the context of T cell effector activities

P5L156: “he immune modulation resulting from ICIs usage can alter…” change to “ The immune modulation resulting from ICIs usage can alter…”

Author Response

REVIEWER 1

This is a review paper titled "Immune-Related Adverse Events due to Cancer Immunotherapy: Immune Mechanisms and Clinical Manifestations".

This review thoroughly analyzes the immunological processes and clinical manifestations of immune-related adverse events (irAEs) linked to cancer immunotherapy with immune checkpoint inhibitors (ICIs). The authors provide a detailed explanation of how immune checkpoints, such as CTLA-4 and the PD-1/PD-L1 axis, play a crucial role in regulating the immune system. The review outlines the different clinical symptoms of irAEs in various organ systems, covers their frequency and risk factors, and highlights the importance of developing better biomarkers and educating clinicians and patients to improve the management of these side effects as the use of ICIs grows in cancer treatment.

authors are requested to:

  • Explore the complex interplay between ICI-induced immune modulation and the triggering or exacerbation of autoimmune conditions.

The Authors answer: Thanks for the comment. A specific and extensive section has been added, that goes under the heading “Host-specific and environmental factors predisposing to irAEs”, Chapter 4, from line 389), as requested. In this chapter, the “Pre-existing autoimmunity” paragraph (line 482) explains in depth the complex relationship between autoimmune conditions and ICIs.

  • Expand the discussion on genetic, and add section on environmental, or patient-specific factors that may influence the risk and severity of irAEs.

The Authors answer: Thanks for the comment. We significantly expanded this section and expanded this topic, as per Reviewer’ suggestion, creating a specific section entitled “Host-specific and environmental factors predisposing to irAEs” (Chapter 4, from line 389).

Overall, the paper is well-written, but there is minor English language errors or typos that can be corrected.

The Authors answer: Thanks for this comment, we reviewed all the paper, we corrected all the typos and rephrased some sentences.

  • Abstract, Line 26 : Change "Here we reviewed and discuss discussed the mechanisms….” To "Here we reviewed and discussed the mechanisms of action…”

The Authors answer: Thanks for highlighting this point, we corrected accordingly. 

  • P4L143-144:correct the grammatical erros and rewrite as follow “The expression of PD1 is a consequence of T and B cell activation and is mediated by the action of inflammatory cytokines, such as IFNγ, in the context of T cell effector activities

The Authors answer: Thanks for highlighting this point, we corrected accordingly.

  • P5L156: “he immune modulation resulting from ICIs usage can alter…” change to “ The immune modulation resulting from ICIs usage can alter…”

The Authors answer: Thanks for highlighting this point, we corrected accordingly.

Reviewer 2 Report

Comments and Suggestions for Authors

This is a comprehensive and well documented report that describes adverse events associated with ICI in the treatment of cancer. The review is fine, but there are TOO MANY errors in grammar and syntax. I have noted the errors and how to fix them for much of the manuscript. The authors should consult a colleague with  English as a first language and fix ALL of the errors.

Comments on the Quality of English Language

There are TOO MANY errors in grammar and syntax. I have noted the errors and how to fix them for much of the manuscript. The authors should consult a colleague with  English as a first language and fix ALL of the errors.

Author Response

REVIEWER 2

This is a comprehensive and well documented report that describes adverse events associated with ICI in the treatment of cancer. The review is fine, but there are TOO MANY errors in grammar and syntax. I have noted the errors and how to fix them for much of the manuscript. The authors should consult a colleague with  English as a first language and fix ALL of the errors. There are TOO MANY errors in grammar and syntax. I have noted the errors and how to fix them for much of the manuscript. The authors should consult a colleague with  English as a first language and fix ALL of the errors.

The Authors answer: Thanks for this comment, we acknowledge that there were a lot of errors, therefore we reviewed all the paper, corrected all the typos and rephrased some sentences.

Reviewer 3 Report

Comments and Suggestions for Authors

The review outlined the pathogenic mechanisms underlying immune-related Adverse Events (irAEs) resulting from the disruption of self-tolerance by Immune Checkpoint Inhibitors (ICIs). It investigated the clinical presentations of irAEs classified by organ involvement, their frequency, and associated risk factors. The review were well -structured and effecively elucidate the key point. However, it contained spelling error and few topics requring further discussion:

1. Is it any literature explain the mechanism why PD1/PDL1 have more irAEs neurological, cardiovasscular, hematologics, muscolosskeletal and rheumatic than CTLA-4 inhibitors. In addition, CTLA-4 inhibitors caused commonly irAEs in endocrinologic, cutaneous and gastrointestinal than PD1/ PDL1 inhibitor.

2. It should be have a table to summarize the onset time and duration of irAEs following adminstration of ICI would be benificial

3. Immune checkpoint inhibitors (ICIs) have proven effective in the treatment of numerous cancers; however, they have been associated with immune-related adverse events (irAEs), among which cytokine release syndrome (CRS) has been reported. The authors should address and discuss about this irAE.

Author Response

REVIEWER 3

The review outlined the pathogenic mechanisms underlying immune-related Adverse Events (irAEs) resulting from the disruption of self-tolerance by Immune Checkpoint Inhibitors (ICIs). It investigated the clinical presentations of irAEs classified by organ involvement, their frequency, and associated risk factors. The review were well -structured and effecively elucidate the key point. However, it contained spelling error and few topics requring further discussion:

  1. Is it any literature explain the mechanism why PD1/PDL1 have more irAEs neurological, cardiovasscular, hematologics, muscolosskeletal and rheumatic than CTLA-4 inhibitors. In addition, CTLA-4 inhibitors caused commonly irAEs in endocrinologic, cutaneous and gastrointestinal than PD1/ PDL1 inhibitor.

The Authors answer: Thanks for this comment. We would like to highlight that the data we showed are taken from meta-analyses pulling together all the available clinical data (see references). That said, there is a combination of factors explaining why some irAEs are more frequent with anti PD1/PDL1 vs. CTLA4 inhibitors. Altogether, we could speculate that this may depend by the interplay between the patient predisposing factors (e.g. immune system alterations, such as pre-existing autoimmune conditions, etc., as described in the new specific section Chapter 4), the differential expression of the antigens targeted by these monoclonal antibodies in the different tissues (e.g. Protein Atlas show that CTLA4 is more frequently expressed in GI as compared to PD1 ligand, for instance), the level at which these two immune checkpoints of T cells act in modulating the immune response (i.e. CTLA4 early, primary in lymph nodes while PD1 later, primarily in cancer tissues), and the frequency of the indications of the different monoclonal antibodies (i.e. PD1 pathway inhibitors collectively are licenced for more cancer types as compared to CTLA4 inhibitors). We added a paragraph to answer the Reviewer (Chapter 5, lines 547-558) and a few sentences of clarification in the Figure 3 legend.

  1. It should be have a table to summarize the onset time and duration of irAEs following adminstration of ICI would be beneficial

The Authors answer: Thanks for this comment. We added this information in a short paragraph, lines 559-563, as per reviewer’s requests. Since the review was already quite long, we preferred not to add this data in another Table.

  1. Immune checkpoint inhibitors (ICIs) have proven effective in the treatment of numerous cancers; however, they have been associated with immune-related adverse events (irAEs), among which cytokine release syndrome (CRS) has been reported. The authors should address and discuss about this irAE.

The Authors answer: Thanks for this comment. The section explaining toxicities related to inflammation and cytokine release has been expanded to address the cytokine release syndrome (Chapter 3.5, from line 354), as requested by reviewer. CRS remains, however, quite rare.

Reviewer 4 Report

Comments and Suggestions for Authors

The manuscript "Immune-Related Adverse Events due to Cancer Immunotherapy: Immune Mechanisms and Clinical Manifestations" reviews the pathogenic mechanisms and clinical manifestations of immune-related adverse events (irAEs) due to immune checkpoint inhibitors (ICIs). It is a comprehensive review of the mechanisms and clinical manifestations of irAE. Overall, the manuscript is well-constructed, providing valuable insights into the mechanisms and manifestations of irAEs in cancer immunotherapy.

The manuscript could gain from a more detailed exploration of the field's current limitations and future prospects, especially in developing strategies to mitigate these adverse effects.

A typo in line 100 should be corrected.

Author Response

REVIEWER 4

The manuscript "Immune-Related Adverse Events due to Cancer Immunotherapy: Immune Mechanisms and Clinical Manifestations" reviews the pathogenic mechanisms and clinical manifestations of immune-related adverse events (irAEs) due to immune checkpoint inhibitors (ICIs). It is a comprehensive review of the mechanisms and clinical manifestations of irAE. Overall, the manuscript is well-constructed, providing valuable insights into the mechanisms and manifestations of irAEs in cancer immunotherapy.

The manuscript could gain from a more detailed exploration of the field's current limitations and future prospects, especially in developing strategies to mitigate these adverse effects.

The Authors answer: Thanks for this comment. We added a paragraph on the current limitations, future perspectives and strategies to mitigate irAE: we acknowledge that this represents a gap in the knowledge, deserving further and in-depth discussion (lines 932-946).

A typo in line 100 should be corrected.

The Authors answer: We could not find any typo in line 100, but we extensively reviewed all the text and corrected all the typos we found.

Reviewer 5 Report

Comments and Suggestions for Authors

The study delves into the mechanisms and clinical implications of immune checkpoint inhibitors (ICIs) and their associated immune-related adverse events (irAEs), highlighting the complex interplay between therapeutic immune activation and autoimmune-like side effects. It underscores the heterogeneity and potential severity of irAEs across various organs, drawing parallels between irAE mechanisms and autoimmune diseases. Moreover, the study emphasizes the need for a deeper understanding of these mechanisms to improve clinical management and patient outcomes in ICI therapy. However, I would like to bring to authors attention some concerns and suggestions for improvement

Section 1

  1. Mechanism of Action Detail: While the section outlines the general mechanism of action of ICIs, it could benefit from a more in-depth discussion, especially regarding the specific checkpoint targets and the downstream effects of blocking these checkpoints. This could enhance the reader's understanding of the immunological processes involved.
  2. Clinical Manifestations of irAEs: The transition to discussing immune-related adverse events (irAEs) is somewhat abrupt. Consider providing a smoother transition or introductory sentence before delving into irAEs. Additionally, the section would benefit from more detailed information on the clinical manifestations of irAEs, perhaps with examples or case studies.

Section 2

  1. The transition between discussing CTLA4 and the PD1/PDL1 axis could be smoother. Consider incorporating a brief transitional sentence to maintain the flow of the narrative.
  2. Clarification on Figure 1: While Figure 1 is informative, a brief caption or description directly in the text could further clarify its purpose and highlight specific elements. This would ensure that readers fully grasp the relevance of the figure to the discussed concepts.

Section 3

    1. Environmental and Genetic Factors: The section mentions the influence of the environment, gut microbiome, and host genetic background on irAEs. It would be valuable to elaborate on these factors, providing examples or evidence to support these claims.
    2. Discussion on Limitations: Acknowledging the limitations of the current understanding of irAEs would add nuance to the discussion. For instance, discussing cases where the mechanisms are not entirely clear or where conflicting evidence exists.
    3. Connection to Clinical Practice: While the section mentions the increasing use of ICIs in clinical practice, it might benefit from a brief discussion of the practical implications for clinicians, such as early detection, management, or monitoring strategies for irAEs.

Section 4

  1. Transition between Sections: Improve the transition between general information on irAE distribution and the detailed breakdown by organ systems. Consider summarizing key points before diving into specific manifestations.
  2. Visual Aids: While Figure 3 is informative, consider adding more figures or tables to aid in visualizing complex data, especially in the later sections discussing specific manifestations. Visuals can enhance reader understanding.
  3. Citations: Ensure that all information, especially statistics and metanalysis results, is properly cited. Readers should be able to trace the information back to its source.
  4. Case Studies: Integrate specific case studies or scenarios illustrating the challenges clinicians face when dealing with irAEs. Real-life examples can enhance the practical understanding of managing these events.
  5. Cohesiveness: Ensure smooth transitions between different aspects of irAEs. For instance, connect the discussion on clinical manifestations with subsequent sections on mechanisms to maintain a cohesive narrative.

Author Response

REVIEWER 5

The study delves into the mechanisms and clinical implications of immune checkpoint inhibitors (ICIs) and their associated immune-related adverse events (irAEs), highlighting the complex interplay between therapeutic immune activation and autoimmune-like side effects. It underscores the heterogeneity and potential severity of irAEs across various organs, drawing parallels between irAE mechanisms and autoimmune diseases. Moreover, the study emphasizes the need for a deeper understanding of these mechanisms to improve clinical management and patient outcomes in ICI therapy. However, I would like to bring to authors attention some concerns and suggestions for improvement

Section 1

  1. Mechanism of Action Detail: While the section outlines the general mechanism of action of ICIs, it could benefit from a more in-depth discussion, especially regarding the specific checkpoint targets and the downstream effects of blocking these checkpoints. This could enhance the reader's understanding of the immunological processes involved.

The Authors answer: we appreciated this suggestion; we added 2 sentences to make this passage smoother (lines 128-1231).   

  1. Clinical Manifestations of irAEs: The transition to discussing immune-related adverse events (irAEs) is somewhat abrupt. Consider providing a smoother transition or introductory sentence before delving into irAEs. Additionally, the section would benefit from more detailed information on the clinical manifestations of irAEs, perhaps with examples or case studies.

The Authors answer: Thanks for this comment. A specific and extensive section has been added discussing “Host-specific and environmental factors predisposing to irAEs” (Chapter 4, from line 389) between the cellular and molecular immunology section – describing the pathogenesis of irAE- and the clinical manifestations’ section. We believe that now the transition between these 2 sections is much smoother. Although we appreciated the hint of adding some case studies, the review is already quite long (> 7000 words), and for this reason we avoided to add other information on clinical examples.

Section 2

  1. The transition between discussing CTLA4 and the PD1/PDL1 axis could be smoother. Consider incorporating a brief transitional sentence to maintain the flow of the narrative.

The Authors answer: we appreciated this suggestion; we added 2 sentences to make this passage smoother (lines 125-128).   

  1. Clarification on Figure 1: While Figure 1 is informative, a brief caption or description directly in the text could further clarify its purpose and highlight specific elements. This would ensure that readers fully grasp the relevance of the figure to the discussed concepts.

The Authors answer: Thanks for this comment. We added a couple of sentences as per Reviewer’s suggestion (lines 138-144) clarifying the purpose of Figure 1, that adds to lines 131-138 that already focused on the PD1 and its expression in tissues and immune cells.

Section 3

    1. Environmental and Genetic Factors: The section mentions the influence of the environment, gut microbiome, and host genetic background on irAEs. It would be valuable to elaborate on these factors, providing examples or evidence to support these claims.

The Authors answer: Thanks for the comment. We significantly expanded the gut microbiota / genetics / host factors’ section, as per Editor’s and Reviewers’ suggestion, and created a specific paragraph entitled “Host-specific and environmental factors predisposing to irAEs”, exploring their contribution to irAEs (Chapter 4, from line 389), providing adequate references.

    1. Discussion on Limitations: Acknowledging the limitations of the current understanding of irAEs would add nuance to the discussion. For instance, discussing cases where the mechanisms are not entirely clear or where conflicting evidence exists.

    1. Connection to Clinical Practice: While the section mentions the increasing use of ICIs in clinical practice, it might benefit from a brief discussion of the practical implications for clinicians, such as early detection, management, or monitoring strategies for irAEs.

The Authors answer: Thanks for the comment. We added a single paragraph discussing the limitations, future perspectives, and the clinical strategies to mitigate irAE – combining the points 2 and 3 raised above by this Reviewer (lines 923-937), we acknowledge that this is interesting and deserve an in-depth discussion. However, for brevity, we decided not to discuss explanatory cases highlighting the gap in the knowledge on irAE (although interesting).

Section 4

  1. Transition between Sections:Improve the transition between general information on irAE distribution and the detailed breakdown by organ systems. Consider summarizing key points before diving into specific manifestations

The Authors answer: Thanks for the comment, we significantly rephrased the text, ensuring higher readability and transitions between different paragraphs.

  1. Visual Aids:While Figure 3 is informative, consider adding more figures or tables to aid in visualizing complex data, especially in the later sections discussing specific manifestations. Visuals can enhance reader understanding.

The Authors answer: Thanks for the comment, please notice that Table 2 summarize the most important points of clinical manifestations, we made the effort to do this, by including a lot of informative data (there are not a lot of reviews that comprehensively summarize irAE mechanisms and clinical manifestations in a single paper).

  1. Citations: Ensure that all information, especially statistics and metanalysis results, is properly cited. Readers should be able to trace the information back to its source.

The Authors answer: We double checked and revised all the bibliography, for a total of 173 references. Although there are a lot of articles on the topic, we believe that we properly cited all the key references.

  1. Case Studies: Integrate specific case studies or scenarios illustrating the challenges clinicians face when dealing with irAEs. Real-life examples can enhance the practical understanding of managing these events.

The Authors answer: As mentioned before, we appreciated the hint, but for brevity we avoided to include real life examples, that would have certainly integrated the reader knowledge, but would have substantially increased the length of this already extensive review (more than 7000 words without clinical scenarios).

  1. Cohesiveness: Ensure smooth transitions between different aspects of irAEs. For instance, connect the discussion on clinical manifestations with subsequent sections on mechanisms to maintain a cohesive narrative.

The Authors answer: Thanks for the comment, we significantly rephrased the text, in some point we believe that the connections are much smoother (e.g. as already mentioned, between the mechanistic/pathogenic paragraphs and the clinical paragraphs, by adding a section on “Host-specific and environmental factors predisposing to irAEs”, Chapter 4).

Round 2

Reviewer 5 Report

Comments and Suggestions for Authors

No further comments to authors